# Use and users of a social science research data archive

**Elina Late** * , **Jaana Kekäläinen**

Faculty of Information Technology and Communication Sciences, Tampere University, Tampere, Finland

These authors contributed equally to this work.
* Elina.Late@tuni.fi

## Abstract

This study focuses on the use and users of Finnish social science research data archive. Study is based on enriched user data of the archive from years 2015–2018. Study investigates the number and type of downloaded datasets, the number of citations for data, the demographics of data downloaders and the purposes data are downloaded for. Datasets were downloaded from the archive 10346 times. Majority of the downloaded datasets are quantitative. Quantitative datasets are also more often cited, but the number of citations vary and does not always correlate with the number of downloads. Use of the archive varies by user's country, organization, and discipline. Datasets from the archive were downloaded most often for study work, bachelor's and master's theses, and research purposes. It is likely that reusing research data will increase in the near future as more data will become available, scholars are more informed about research data management, and data citation practices are established.

## Introduction

The openness of research should be self-evident by the very nature of scientific inquiry relying on public criticism. Obviously, this is not the whole truth because there has been a need for an open science movement [1, 2]. Recently, the questions related to open science, open research data and open access have been lively discussed among research communities and policy makers (see e.g. [3–5]). The FAIR data principles [6, 7] are commonly accepted, and lately even criticized for not being sufficient [8]. These ideas have also affected research processes, not least because of the requirements by funding bodies (see e.g. [9–11]). Openness and sharing are becoming important factors in the evaluation of impact, whether it concerns research infrastructures or scholars (see e.g. [12]).

Sharing research data is an essential aspect in open science because of the possibility to verify given results and to enhance the effectiveness of research by the reuse of data. For this purpose, a great number of repositories has been established for different disciplines, and across disciplines, like European open science cloud [13]. The varying characteristics of disciplines cause differences in research data deposing and reuse practices, which affect the organization of open research data repositories [14, 15]. In physical and life sciences, and branches of medicine, the need for research data is immense, and thus the reuse seems profitable. There are

**Data Availability Statement:** Data are available from the Finnish Social Science Data Archive: (https://services.fsd.uta.fi/catalogue/FSD3424?tab=description&study_language=en&lang=en).

**Funding:** The author(s) received no specific funding for this work.

**Competing interests:** The authors have declared that no competing interests exist.

many open research data repositories for these fields (see e.g. [16–18]). In humanities and social sciences, sharing research data has not been as prevalent but research data repositories exist as well (see e.g. [19–22]). Besides research data repositories and databases, research journals have also started to publish research data pertaining to published articles. These data, however, are rather for verification than reuse purposes.

Release of research data as well as the organization of repositories have drawn a lot of attention, but are the opened data reused? The type of research data attracting interest for potential reuse and the purposes of reuse are less studied, especially concerning open social science research data. This information is vital to understand the evolving knowledge creating practices, impact of research and the development of open science. We focus on these issues by analyzing usage data from Finnish Social Science Data Archive (FSD). We set forth the following research questions:

1. How many times data were downloaded from the FSD archive during 2015–2018?

2. What types of data were downloaded from the FSD archive?

3. How many times were the most downloaded data cited?

4. What organizations, disciplines, and countries the users of data represent?

5. For what purposes were data downloaded?

In Section 2, we review the earlier literature pertaining to sharing and reusing research data. In Section 3, our user data and research methods are introduced. Results are presented in Section 4, they are further discussed in Section 5 and Section 6 concludes the article.

## Literature review

### Research data sharing

Sharing research data demands infrastructure for data management. The research data vary greatly by disciplines, yet there is no clear-cut division into quantitative and qualitative disciplines in the era of digitalization. Nevertheless, different data types need different solutions for storage and access. The question is not only about the data formats, like numeric or textual data, but also about the ownership of data and rights to use them. The data practices and research methods of disciplines also affect the management solutions. The infrastructure for research data includes, among others, repositories, databanks, data grids, databases, archives and digital libraries. The infrastructure has been described and discussed in several articles (for *social sciences and humanities* [23–25]; for *natural sciences* [6, 26–29]).

Legal and ethical issues are to be considered in data sharing. Participants in empirical research need to give informed consents for data reuse; data may contain sensitive information and thus need anonymization. Proprietary rights, copyrights and commercial interests are often involved. Data management needs planning and appropriate metadata. [30–32] The creation and/or collection context of the data, the purpose of data creation/collection, storage format and access rights are essential information for the reuse of data. [33, 34] Providing such metadata demands expertise on data management, knowledge about the data and the context of their usage. All this means that sharing data involves costs.

Researchers' data practices and attitudes are crucial for sharing research data. The first prerequisite is awareness of the possibilities of sharing and infrastructure. Data practices change towards openness as funding bodies and scholarly journals require data sharing and open publishing. Pampel and Dallmeier-Tisel [30] emphasize the effect of incentives. Researchers themselves have started to insist opening research data for verification and replication purposes

(e.g. [2, 35]). Nevertheless, researchers may also have negative attitudes towards data sharing. Concerns about misuse, misinterpretation, lack of confidentiality and loss of intellectual property are typical [36, 37]. Ethical issues, lack of funding, time or knowledge about the possibilities are also mentioned as barriers to data sharing [30, 38].

Kim and Adler [39] formed a survey-based model of social scientists' data sharing behavior. Perceived career benefit and normative influence were the most important factors with a positive effect on data sharing behavior. Perceived effort and career risk were the principal factors with negative effect on data sharing behavior in the model. Chawinga and Zinn [40] conducted an intensive literature review on research data sharing, based on 105 research papers. They analyzed factors affecting research data sharing at individual, institutional and international levels. Besides aforementioned reasons, they mention that at individual level experienced researchers are more willing to share data than early career researchers are. At institutional level, the main driving forces and hindrances are (lack of) training in research data sharing, compensation and institutional policies. At international level, they mention research funding agencies' policies, publishers' policies, infrastructure for research data management, and rights management.

## Research data reuse

Use and reuse of research data is an essential distinction. The former refers to the use of data collected from primary sources for the purpose and project they were originally aimed for; the latter refers to the use of data from secondary sources, or data originally collected for other purposes than the current use. [41] Repositories consisting of datasets deposited by researchers aim to enhance data reuse. These are of interest for our study.

What do we know about the reuse of open research data? Reuse has some prerequisites [1, 41, 42]:

- the data must be findable, which requires informative metadata providing context for the data

- the data usage licensing must be applicable and clear

- reuse might demand the integration of data with other data, or transformation of the format to be compatible with analysis methods, which sets requirements for the data formats.

Data use and reuse has often been studied with interviews and surveys (e.g. [15, 37, 38, 43, 44]). According to these studies, data reuse is not very extensive.

Next, we review studies focusing on the reuse of research data in social sciences. Again, most studies concentrate on researchers' attitudes and practices investigated through surveys or interviews (e.g. [45–50]). The reuse of quantitative data probably is more common than reuse of qualitative data because the number of opened quantitative datasets is greater [51] and metadata for quantitative data are easier to produce. Nevertheless, studies on the reuse of qualitative data in social sciences are quite numerous.

Yoon [49] examined reusing failures with 23 quantitative social science researchers. The main reasons for failures were wrong or incomplete description of data, difficult access to data, lack of interoperability in data formats and software, and missing values or improper manipulation of data. Faniel and others [45] conducted a survey on satisfaction with data reuse involving 237 respondents from social sciences. Their analysis revealed five constructs affecting researchers' data reuse satisfaction: data completeness, data accessibility, ease of operation, data credibility, and documentation quality. Curty [51] explored factors influencing research data reuse in social sciences through a survey involving 564 participants. Also Curty mentions

data documentation, data fitness, producer trustworthiness and credibility, data quality and study rigor as the most important factors influencing social scientists' data reuse. These results are all in line.

Qualitative data reuse is studied by Bishop and Kuula-Luumi [52]. They utilized two data sources: 1) the downloads of qualitative data from two data repositories, UK Data Service and FSD, 2) citations to qualitative data in scientific publications during 1990–2015 obtained from the Web of Science. The data from UK data service consist of 7,155 downloads of 267 datasets in the years 2002–2014. The FSD data include 550 downloads in the years 2014–16, the number of datasets is not mentioned. Their analysis reveals also user groups downloading data and their purposes for the data reuse. In UK Data Service, the three most frequent user groups were postgraduates (41.7%), staff at institutes of higher education (26.9%) and undergraduates (25%). Other users include other students, other staff, and commercial users and others. The typical reuse purposes correspond to the biggest user groups: learning (63%), research (15%) and teaching (13.4%)–rest are miscellaneous purposes. For FSD, the user groups are not mentioned but the reuse falls in four categories: studying (41%), master's theses (28%), teaching (20%) and research (11%).

The results concerning publications doing secondary analysis or reusing qualitative data are not connected to the downloaded datasets. Bishop's and Kuula-Luumi's [52] analysis shows that either reuse is not very common or research data are not cited; the total number of citing articles is 347 over 25 years. However, it seems that number of citing publications is increasing.

The current study is an extension to Bishop's and Kuula-Luumi's contribution. We also seek to explore reuse through download and user data but with a new dataset including both qualitative and quantitative datasets. This kind of analysis enriches the results gained through surveys and interviews with realized actions on data. Nevertheless, downloading does not always entail reuse. We complement our study with an analysis of citations to most downloaded datasets.

## Research data and methods

Finnish Social Science Data Archive was founded in 1999 for archiving, promoting and disseminating digital research data for research, teaching and learning. The archive is a unit of Tampere University with responsibility to serve as a national resource center for social science research. FSD is also a co-operator of Consortium of European Social Science Data Archives (CESSDA) and has the Core trust seal certification [53]. FSD offers researchers data curation services, like the description of data, selection of file formats suitable for long-term reservation and reuse, anonymization. A web search portal serves findability with several search facets. All services are free of charge. (See [54])

Currently (17.12.2019), the archive contains 1494 datasets. The data are both quantitative (1266 datasets, 85%) and qualitative (228, 15%). Mainly, the data are in Finnish but there are several datasets in English (381) and a few in Swedish (20). The availability of the datasets divides into four categories: a) openly available for all users without registration, b) available for research, teaching and study, c) available for research only (including master's and doctoral theses, d) available only by permission from the data depositor/creator. The depositor of the data can decide on which terms the dataset can be downloaded. There are about 3200 registered users in FSD.

Primary research data used in this study were collected by the FSD [55]. The data consist of quantitative user data of the archive during 2015–2018. The data contain the number of downloaded datasets by year and month, the identification numbers and names of the downloaded

data, the quality of the downloaded data (qualitative/quantitative), the availability and use terms of the downloaded data.

The data are enriched with information about the users and the use of the data. Each time a person downloads data from the archive (availability categories b-d) she/he is asked to answer to certain questions. Users are asked to indicate their institution, discipline, country, and the reason data are downloaded for. Unfortunately, information about users discipline and country is not available for every download since it is possible to download certain type of data without registration. These data are open data that are available for any use. This is why we do not have information about users institution, discipline and country for every downloaded dataset. Thus, the number of cases (N) varies in our data between 7566–10346.

The data for this study were gathered into one sav-file and analyzed with SPSS program. The data are analyzed by quantitative methods using frequencies and crosstabs. Statistical significance is tested with $chi^2$ test.

Ten most frequently downloaded quantitative and qualitative datasets were analyzed in more detail. For these datasets the type of data (for example survey, statistics, interviews etc.) was traced from the FSD database. Furthermore, we collected citations to these datasets from the years 2015–2018. Citations were collected first from the FSD database. FSD asks the downloaders of the data to inform the archive of any publication the data are used for. It should be noted that all of these publications do not formally cite datasets although data has been used in the study. Citations were retrieved also for each dataset from Google Scholar and from the Web of Science. Each document found from Google Scholar or Web of Science was checked to ensure that the dataset was actually used in the publication. Citations were collected into one excel file and the type of citing document (from bachelor's and master's theses, doctoral dissertations, research publication) was identified.

## Findings

### Frequency of downloading data

Datasets were downloaded from the archive 10346 times during 2015–2018 (Fig 1). More than 2000 datasets were downloaded from the archive each year. The number of downloads increased from 2015 to 2016–2017 by 28%. However, for some reason the number of downloads decreased by 19% in 2018 from the number of downloads in 2017. Further increase would have been expected since the number of deposited datasets increase every year. The peak in the number of downloads happened in the fall as data were downloaded most frequently in November. Downloading data was less frequent during the summer months June, July and August.

A total of 1039 individual datasets were downloaded from the archive during 2015–2018. Most commonly datasets were downloaded only once (25.6%) or twice (16.8%). However, there is also a small number of heavily downloaded datasets in the archive. Nine datasets were downloaded more than 100 times. One survey dataset was downloaded 630 times during the four-year period.

### The type of downloaded data

As most of the data deposited in the archive are quantitative so are most (85.4%) of the downloaded datasets (Fig 1). Downloading quantitative datasets increased from 2015 to 2017 by 27%. However, the number of downloads decreased from 2017 to 2018 by 21%. Typically, a quantitative dataset is survey data. The 10 most frequently downloaded quantitative datasets were studied more carefully (Table 1). All of these datasets were downloaded at least 100 times. For example the most heavily downloaded dataset "Measures of Democracy 1810–2012" (630

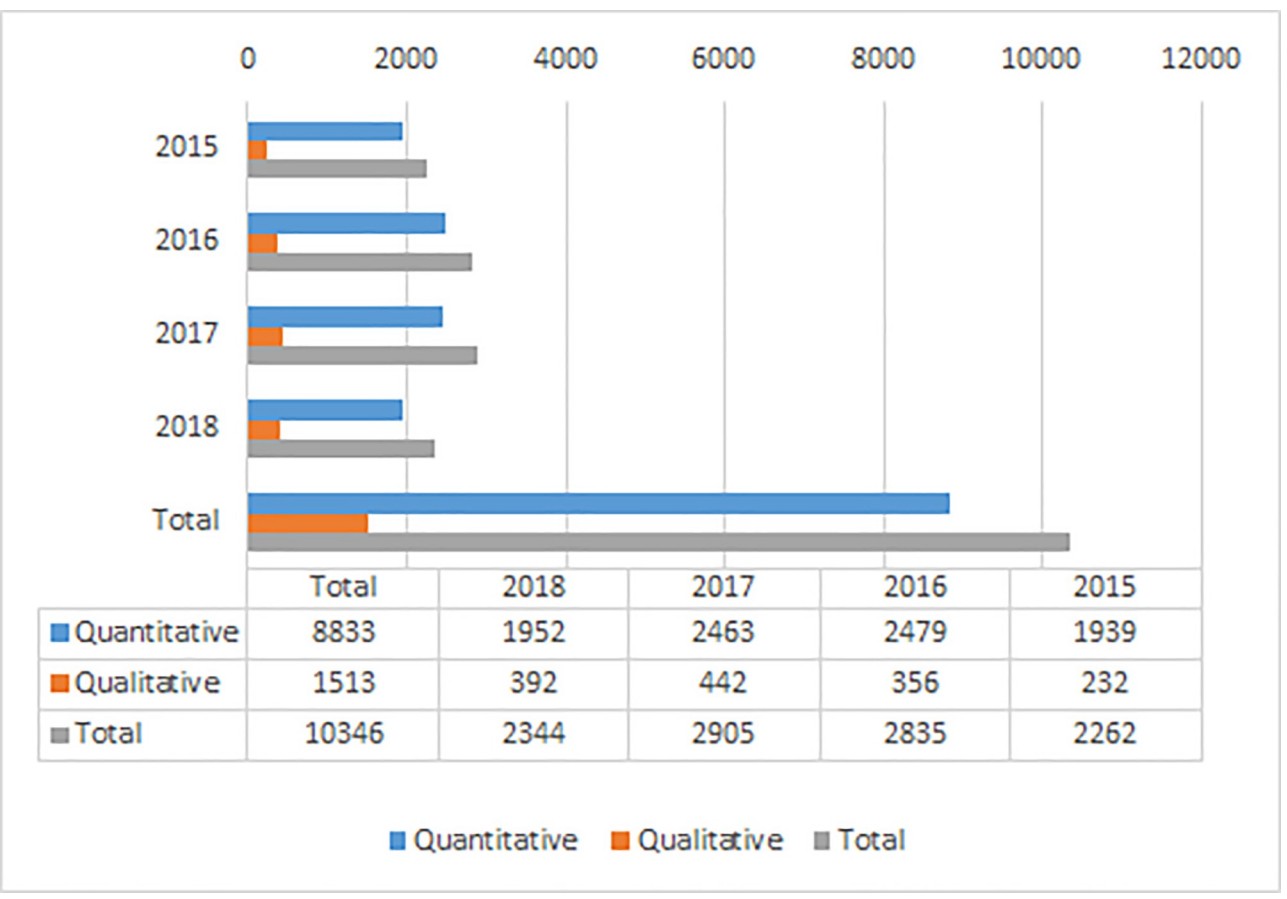

**Fig 1. Number of downloads of quantitative and qualitative datasets from FSD during 2015–2018.**

downloads) is part of a group of datasets by professor Vanhanen and provides comparable data on the degree of democratization in nearly all independent countries of the world from 1810 to 2012. In addition, especially large international longitudinal survey datasets such as"European Social Survey" are downloaded frequently. Also data from large national longitudinal studies, such as "EVA Survey on Finnish Values and Attitudes"or "Finnish National Election Study" are frequently downloaded.

**Table 1. Names, type, and number of downloads and citations for 10 most frequently downloaded quantitative datasets from the repository 2015–2018.**

|  | Type of data | Number of downloads | Number of citations |
|---|---|---|---|
| 1. Measures of Democracy 1810–2012 | Statistics | 630 | 40 |
| 2. Finnish Lifestyles Survey 1995 | Survey | 418 | - |
| 3. Democratization and Power Resources 1850–2000 | Statistics | 248 | 41 |
| 4. European Social Survey 2014: Finnish Data | Survey | 205 | 2 |
| 5. Index of Power Resources (IPR) 2007 | Statistics | 172 | 12 |
| 6. Finnish National Election Study 2015 | Survey | 160 | 56 |
| 7. Survey on Finnish Values and Attitudes 2015 | Survey | 117 | 5 |
| 8. EVA Survey on Finnish Values and Attitudes 2016 | Survey | 113 | 5 |
| 9. Finnish National Election Study 2011 | Survey | 110 | 55 |
| 10. Finnish Attitudes to Immigration: Suomen Kuvalehti Survey 2015 | Survey | 100 | 1 |

The number of citations to the most downloaded quantitative datasets varied quite much (Table 1). Slightly surprisingly, the most cited datasets were two surveys, both related to national elections (Table 1, items 6 and 9). Although the theme is national, FSD provides detailed codebooks in English. Most of these citations (102) are from articles published in journals or monographs. Altogether nine citations are from master's or bachelor's theses. Most of the citing authors are from Finland, yet there are a few representing other countries. Two statistics by Prof. Vanhanen are much cited as well (Table 1, items 1 and 3): altogether 81 citations, of which eight are from theses or doctoral dissertations, others from scholarly articles or monographs. The authors citing these statistics represent a much broader variety of countries, probably because the theme of the data is more international. Obviously, the number of citations does not correspond to the number of downloads as all downloads do not lead to use, all use does not realize in publications, and all publications do not cite the used research data.

Almost 15% of the downloads from the archive were for qualitative data (Fig 1). Downloading qualitative data increased from the year 2015 to 2017 by 37%. The number of downloads for qualitative data decreased from 2017 to 2018 by 11%. Qualitative data deposited in the archive are typically in text form such as transcriptions of interviews. The most frequently downloaded qualitative datasets were analyzed in more detail. These datasets were downloaded at least 30 times during 2015–2018. Seven out of the ten most downloaded qualitative datasets are data from writing competitions (Table 2). For example data "Parenthood after Divorce 2011–2012" containing writings from divorced parents were downloaded 94 times. Other frequently downloaded qualitative datasets are for example "Everyday Experiences of Poverty: Study, Research and Teaching Material 2006" and "My Well-being 2010: Writing Competition" both data collected in a writing competition. For example, the Finnish Literature Society organize frequently writing competitions to collect citizens recollections. Dataset "Tales and Stories Told by Children 1995–2005" was downloaded 43 times. This dataset contains the transcriptions of audio recorded stories told by children of different ages.

The number of citations to qualitative datasets varied between 1 to 20 during 2015–2018 (Table 2). In total 10 most downloaded datasets were cited 78 times during the four-year period. Majority (83.3%) of the citations are from bachelor's and master's theses. The rest (16.7%) are from other research publications (journal articles, monographs, research reports). All of the citing authors are from Finland. Most cited dataset was "Everyday Experiences of Poverty 2006" (Table 2, item 6). This dataset was used in 16 bachelor's and master's theses and four research publications. The follow-up data for the same study "Everyday Experiences of Poverty 2012" was cited 12 times (Table 2, item 5). Most of the citations for qualitative dataset were found from the FSD database. Some citations were found from Google Scholar but not once from Web of Science.

**Table 2. Names, type, and number of downloads and citations for 10 most frequently downloaded qualitative datasets from the repository 2015–2018.**

| | Type data | Number of downloads | Number of citations |
|---|---|---|---|
| 1. Parenthood after Divorce 2011–2012 | Writing competition | 94 | 9 |
| 2. My Well-being 2010: Writing Competition | Writing competition | 53 | 9 |
| 3. Everyday Experiences of Poverty: Study, Research and Teaching Material 2006 | Writing competition | 52 | 9 |
| 4. Tales and Stories Told by Children 1995–2005 | Stories written by children | 43 | 5 |
| 5. Everyday Experiences of Poverty 2012: Follow-up Study | Writing competition | 42 | 12 |
| 6. Everyday Experiences of Poverty 2006 | Writing competition | 39 | 20 |
| 7. Social Class of University Students 2009–2010 | Students writings | 36 | 3 |
| 8. Occupational Identities of Fixed-Term Employees 2006 | Interviews | 34 | 6 |
| 9. Cultural Heritage of Finland 2014 | Open ended answers from a survey | 30 | 1 |
| 10. Life Stories of Adults with Cerebral Palsy 2008 | Writings/Narratives | 30 | 4 |

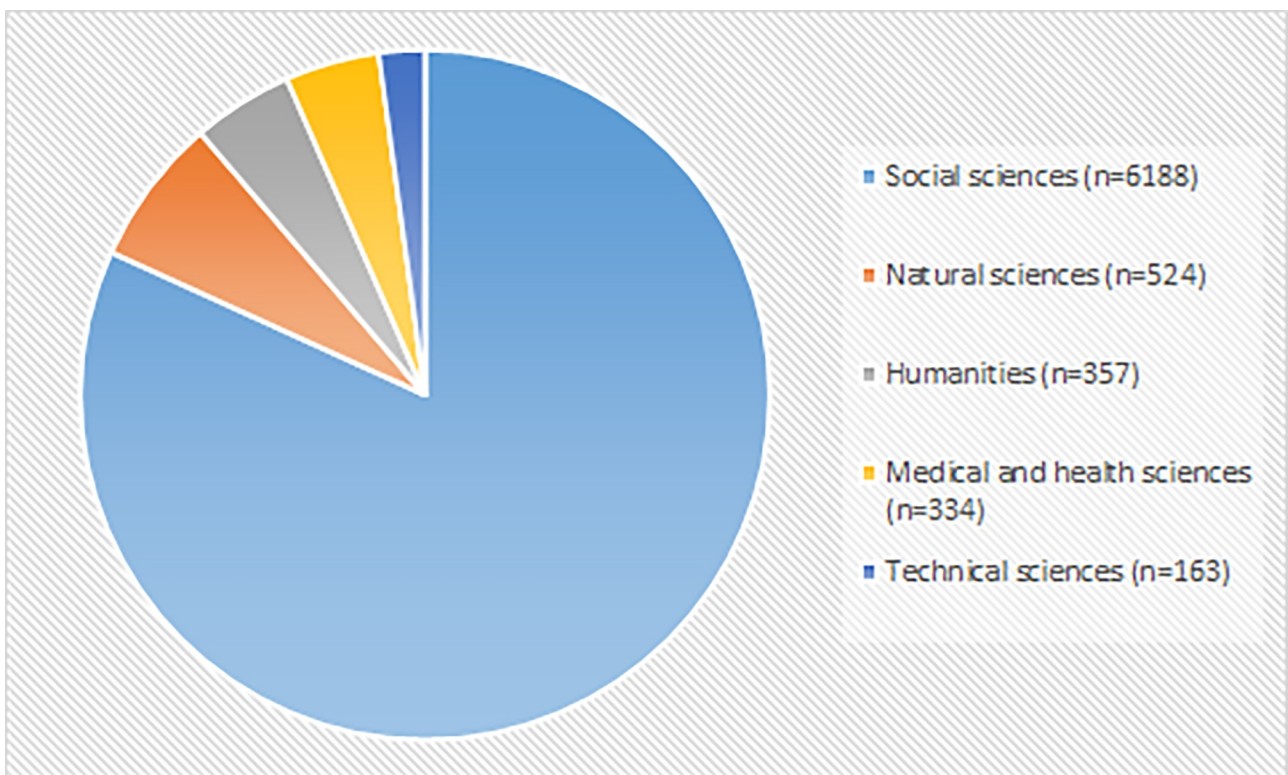

**Fig 2. Discipline of the downloaders of the data (N = 7566).** Information about discipline is missing from 2780 cases.

## Users of the archive

The most typical user downloading data from the archive comes from Finland (88.7% of users), works or studies in a Finnish university (75.5%) and represents social sciences (81.8%). However, there are users from other European countries (for example Germany 334, United Kingdom 82, Romania 51, France 33, Hungary 33, Sweden 32, Denmark 30) and from other parts of the world (for example United States 85, Japan 32, Canada 31).

In addition to universities, users represent other type of organizations such as Finnish universities of applied sciences (10.2%) and foreign universities and research institutes (12.0%). Universities of applied sciences offer teaching mainly on bachelor level and are less focused on research compared with universities. In addition to social sciences, the archive has users representing natural, medical, and technical sciences and humanities (Fig 2). There are some differences between disciplines in the represented organization types (chi .000). Majority of users representing social sciences (91.8%), humanities (95.5%), and natural science (79.0%) work or study in Finnish universities. In addition, majority of the users representing technical sciences (70.2%) and medical and health sciences (52.5%) work or study in Finnish universities of applied sciences.

Although, quantitative datasets are most frequently downloaded in all disciplines, there are differences between the disciplines in the share of downloaded quantitative and qualitative data (p < .000). Users representing medical and health sciences, social sciences, and humanities download more often qualitative data compared with users representing natural and technical sciences (Table 3). Approximately 20–30% of downloads by the users representing medical and health sciences, social sciences, and humanities are for qualitative data. In natural and technical sciences only 1–3% of downloaded data are for qualitative datasets.

**Table 3. Discipline of the users downloading qualitative and quantitative datasets.**

|  | Qualitative | Quantitative |
|---|---|---|
| Natural sciences (n = 524) | 3.2% | 96.8% |
| Technical sciences (n = 163) | 1.2% | 98.8% |
| Medical and health sciences (n = 334) | 33.8% | 66.2% |
| Social sciences (n = 6188) | 19.0% | 81.0% |
| Humanities (n = 357) | 25.2% | 74.8% |
| Total (N = 7566) | 18.5% | 81.5% |

p < .000, Information about discipline is missing from 2780 cases

In addition, there is a clear difference (p < .000) between users from Finland and users from other countries in the share of downloading quantitative and qualitative data (Table 4). It is natural that users from Finland download qualitative data more (20.4% of total downloads) often compared with users from other countries since qualitative data are mainly in Finnish. Many of the quantitative datasets or the codebooks for the datasets are available in English (for example "Measures of Democracy 1810–2012"). Therefore, users outside Finland can often easily use quantitative data from the archive.

In social sciences nine out of ten most frequently downloaded datasets are survey datasets including the "Life style study 1995" downloaded 406 times (6.6% of total n = 6188). Also, national longitudinal surveys for example "Finnish National Election Study" from the years 2011 and 2015 gain lot of downloads by users representing social sciences.

But surprisingly, the greatest share of downloading qualitative data is for those representing medical and health sciences. Seven out of 11 most downloaded datasets by those representing medical and health sciences are in fact qualitative. Most frequently downloaded qualitative datasets are open ended questions separated from large national surveys such as "Living with Depression" and "Parenthood and Alcohol Use". However, the most frequently downloaded dataset by users representing medical and health sciences is a survey dataset "Family, Parenthood, Children's Well-Being and Risks of Exclusion" (18.3% of total downloads n = 334).

In humanities, most frequently downloaded datasets are also surveys. Dataset"Finnish Science Barometer 2013" (3.4% of total n = 357) was downloaded most frequently. However, in the top ten list in humanities are also diary data and interview data related to historical topics such as "Women and War" and "Father-Son Relationships and the War".

If looking at the top ten list of downloaded datasets in natural and technical sciences, all datasets are survey datasets. In natural sciences five out of ten most downloaded datasets are international surveys such as "ISSP 2011: Health, Finnish data" (8.4% of total n = 524). In technical sciences the most downloaded datasets are national surveys, such as "Elderly People and Technology" (10.4% of total n = 163).

**Table 4. Country of users downloading qualitative and quantitative datasets.** Information about country is missing from 2362 users.

|  | Qualitative | Quantitative |
|---|---|---|
| Finland (n = 7079) | 20.4% | 79.6% |
| Other European countries (n = 705) | 2.6% | 97.4% |
| Other countries (n = 200) | 7.0% | 93.0% |
| Total (N = 7984) | 18.5% | 81.5% |

p < .000, Information about country is missing from 2362 cases

**Table 5. Share of downloading quantitative and qualitative (%) data by purpose of downloading.**

|  | Qualitative % | Quantitative % | Total % |
|---|---|---|---|
| Study work / essays (n = 4040) | 12.5 | 87.5 | 39.0 |
| Bachelor's and master's theses (n = 2247) | 27.1 | 72.9 | 21.7 |
| Doctoral theses (n = 513) | 7.6 | 92.4 | 5.0 |
| Research (n = 2286) | 5.5 | 94.5 | 22.1 |
| Teaching (n = 957) | 23.2 | 76.8 | 9.2 |
| Other purposes (n = 303) | 4.0 | 96.0 | 2.9 |
| Total (n = 10346) | 14.6 | 85.4 | 100.0 |

p < .000

## Purposes for downloading data

When downloading the data users are asked for which purpose the dataset is downloaded for. According to this information, the datasets were downloaded most commonly for study work/ essays (39.0%), bachelor's and master's theses (21.7%), and for research purposes (22.1%) (Table 5). Only about 9% of the downloads were made for teaching purposes, 5% for doctoral theses, and about 3% for other purposes. There is a statistical difference (p < .000) in the share of downloaded quantitative and qualitative data according to the purpose the data are downloaded for (Table 5). In all cases, quantitative data are downloaded most often. More than 90% of downloads for the purposes of doctoral theses and research focused on quantitative data. However, for the purposes of bachelor's and master's theses and teaching, qualitative data are downloaded more often compared with other purposes.

There is a variation (p < .000) between disciplines in the share of downloading quantitative and qualitative data (Fig 3). Compared with other disciplines the share of downloading data for study work/essay purposes is greatest for those representing technical sciences (61.3% of total n = 163) and medical and health sciences (60.5% of total n = 334). In addition, compared with other disciplines the share of downloading data for research purposes is greatest for those representing technical sciences (26.4% of total n = 163). The share of downloading data for doctoral theses is greatest for users representing humanities (19.9% of total n = 357).

Purposes for downloading data vary also between the country that user represents (p < .000). When users from Finland download data especially for study work (45.1% of total n = 7079), users from other parts of the world download data mostly for research (71.6% of total n = 905). Surprisingly the share of downloading data for the purposes of doctoral theses is greatest for those from outside Europe. Furthermore, users from Finnish universities and universities of applied sciences download data most often for study work/essays. Users from Finnish research institutes and foreign universities and research institutes download data mainly for research purposes.

Nine out of the ten most downloaded datasets for study work/essays are quantitative survey datasets. The "Life Style study 1995" survey data was downloaded 328 times (8.1% of total n = 4040) for purposes of study work/essay. The same dataset was downloaded also most often for teaching purposes (9.2% of total n = 957). This dataset has been used as an example data in a research methods courses in Tampere University, which most likely explain the high numbers in downloads but low in the number in citations. Statistical dataset "Measures of Democracy 1810–2012" was also in the top ten lists of both study work/essay and teaching purposes which indicates that dataset has been used as a course material.

The most common purpose for downloading qualitative datasets were for bachelor's and master's theses. However, survey datasets dominate the top ten data list of datasets

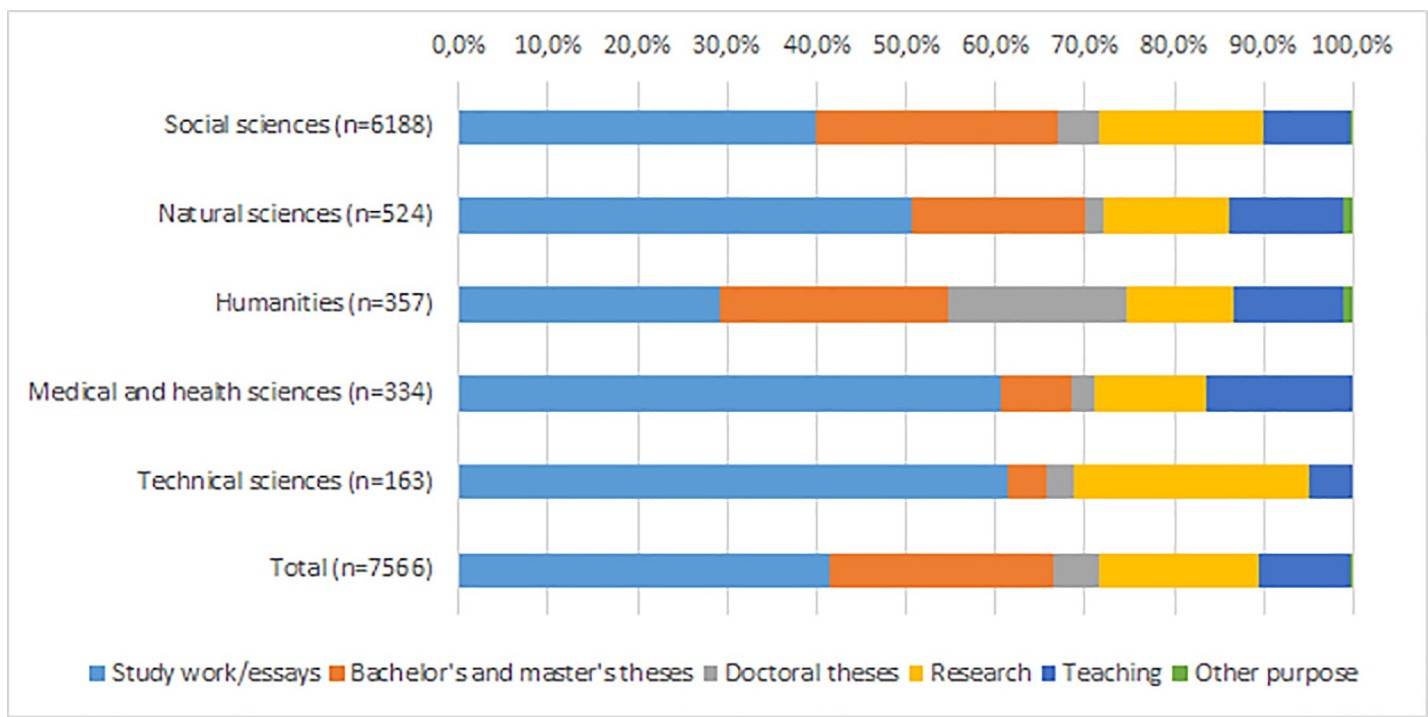

**Fig 3. Share of downloading datasets for different purposes by discipline.** Information about discipline is missing from 2780 cases.

downloaded for the theses. Once again the "Measures of Democracy 1810–2012" dataset was most frequently downloaded (3.6% of total n = 2247). Nevertheless, qualitative datasets from writing competitions such as "Everyday life experiences about poverty" from years 2006 and 2012 are frequently downloaded for bachelor's and master's theses.

If looking at the data downloaded most often for doctoral theses and research purposes, a group of datasets stand out. Three of the most frequently downloaded datasets (Measures of Democracy 1810–2012, Democratization and Power Resources 1850–2000, Index of Power Resources 2007) for doctoral theses and research purposes are parts of the same data collection from the same author (Prof. Vanhanen). Combined downloads for these datasets form 23.1% of the total downloads for research purposes and 18.5% for doctoral theses purposes. In addition Finnish national election studies from years 1991–2015 are frequently downloaded for both doctoral theses and other research work.

For other purposes users download mainly quantitative data such as "Public Procurement Notices" from the years 2011–2017 (38.9% of total n = 303). Public procurement notices data contain public contract notices such as supply, service or public works contracts into which for example the state and municipalities enter with external suppliers. FSD collections contain data beginning from the year 2007.

When depositing data to the archive, donator can decide on which terms and for which purposes the dataset can be downloaded. Most commonly (65.5%) downloaded data were deposit for the use of research, teaching and study work (Fig 4). More than one fourth of the downloaded data were deposit for any use. For example, the most frequently downloaded dataset (630 downloads) "Measures of Democracy 1810–2012" was deposited for any use without registration. Only 5% was deposited only for the use of research and theses. Almost three percentages of the downloadings required permission for use from the donator. Permission from the donator is required more often for qualitative data (6.6%) compared with quantitative data

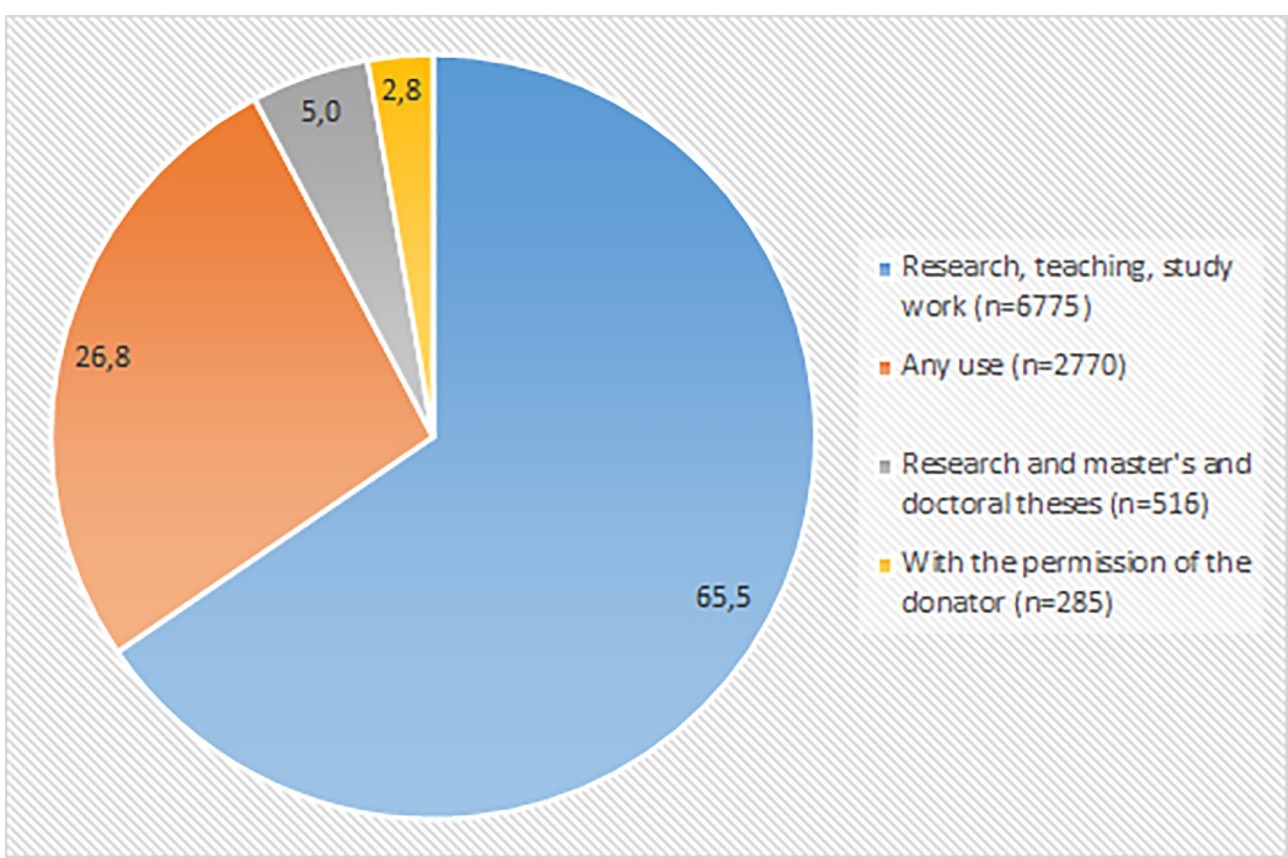

**Fig 4. Terms of use of the downloaded data (N = 10346).**

(2.1%). Datasets requiring permission from the donator are often about sensitive topics such as health issues. For example, the use of datasets "Community-based Mental Health Rehabilitation of Young Adults 2010–2012: Client Interviews" and "University Student Health Survey 2012" require permission. Also the interview data of the frequently downloaded "Everyday Experiences of Poverty" dataset require permission from the donator.

## Discussion

This study focuses on the use and users of a social science research data archive. To our knowledge, this was among the first attempts to study the use and users of social science research data repositories based on user data such as log data. Earlier, data use and reuse have been studied with interviews and surveys (e.g. [15, 38, 43]). First, the number of downloads and second, the type of downloaded data from the archive during 2015–2018 were investigated. More than 2000 datasets were downloaded from the archive each year (a total of 10346 downloads). Downloads focused on 1039 individual datasets and most commonly datasets were downloaded only once or twice.

Clear majority of the downloaded data are quantitative. Result is in line with earlier studies [51]. Especially large international statistics and national longitudinal survey datasets gain lot of downloads. Bishop and Kuula-Luumi [52] studied the number of downloads for qualitative datasets from FSD during 2014–2016. According to their study qualitative datasets were downloaded 550 times. Compared with their findings, our results shows increase in downloads

since qualitative datasets were downloaded 1513 times during 2015–2018. This might indicate growing awareness about research data sharing and benefits of data reuse [39, 40].

Third, the number of citations to the most downloaded quantitative and qualitative datasets were investigated. Our findings show that number of citations varies quite much and the most downloaded datasets are not the most cited. Further, quantitative data are more often cited in comparison with qualitative data. Most citations are from Finnish authors and great part of the citations are from bachelor's and master's theses. It is obvious that citations reveal only a fraction of the use of the data. Citing practices for research data are clearly evolving. Citations are often incomplete or erroneous, yet the more recent they are, the better they are. It seems that all reused research data are not cited, at least not in a formal way. Also, the frequency of citations to reused research data seems to be increasing compared with Bishop's and Kuula-Luumi's study [52]. They reported of 347 citing articles over 25 years. According to our findings, the most ten most downloaded quantitative and qualitative datasets gathered 295 citations in four years. We do not have information about how many of the datasets in our study are published in data journals [56].

Fourth, the users' country, organization and discipline were investigated. The most typical user downloading data from the archive comes from Finland, works or studies in a Finnish university, and represents social sciences. However, there are users from other European countries and outside Europe, other organizations such as universities of applied sciences and from all major disciplines. Users from natural and technical sciences download mainly quantitative data. Users representing medical and health sciences, social sciences, and humanities download also qualitative data.

Although FSD provides search interface in Finnish and English, most of the datasets are in Finnish. However, codebooks are often available also in English. Results show that quantitative data are downloaded from all parts of the world, but qualitative data are downloaded mainly by users from Finland.

Fifth, the purposes for downloading datasets were studied. The most common purposes were study work/essays, bachelor's and master's theses, and research purposes. Using data for teaching and for doctoral theses was less common. Our findings are in line with results offered by Bishop and Kuula-Luumi [52] for purposes downloading qualitative datasets. The user demographics correlate with the purposes for downloading data: Finnish users downloaded data especially for study work and users from other countries download data mostly for research.

The limitations of a log-based study are obvious: downloading is no guarantee of use, and the purpose of downloading given by the users is not necessarily adequate. On the one hand, users may download data that they never use; on the other hand, users may utilize once downloaded data many times and for many purposes. Further, we studied only one archive as an illustrative case. The generalizability of the findings is limited. We studied the citations only for the ten most downloaded quantitative and qualitative datasets. To get broader picture of the citing practices, and for example about the evolution of the number of citations, a further study would be needed.

## Conclusions

This study illuminates the use and users of a social science research data archive. Our results show that the archive is actively used, especially for the needs of education. Although the research data archive investigated in this study focuses on social sciences, users represent all major disciplines. It is also notable, that although most of the deposited datasets are in Finnish,

users represent countries outside Finland and outside Europe as well. Thus, the location or the description of the archive does not entirely define its use and users.

Since research data sharing, reusing and citing is still evolving in social sciences, the topic needs more research. Future will show, if research data will form as a similar research output and merit for scholars as scholarly publications. If so, the citing practices in different disciplines need to be formalised. This will require close monitoring. Research tradition in bibliometrics offer a methodological ground for data citation analyses. Nevertheless, limitations such database coverage and disciplinary practices in knowledge production and citing must be taken into account. Increase in deposited research datasets is expected, which also puts pressure for research data repositories for giving service for data donators. Experts in data management are needed and recruited more in the future. Sharing and deposing data involves costs that need to be taken into account. What is the role of AI in research data management and data repositories and will research data depositing become a commercial business is to be seen.

Furthermore, it would be interesting to study the development of data donators and data types. Depositing qualitative data is still in its infancy and there is no clear picture of in what ways qualitative data are and can be reused. In addition, when data are created more and more outside academia it is important to recognize data practices in different disciplines. It is also important to study, in which format data should be offered to meet the practices and needs of researchers. Further, it is evident to recognize what it demands from the repositories to handle and curate data not primarily collected for research purposes. In order to achieve a whole picture of the research practices and data use both quantitative and qualitative approaches are needed.

## Acknowledgments

This study was made in collaboration with Finnish social science data archive. We thank Hannele Keckman-Koivuniemi and Helena Laaksonen for offering the data for this study and the FIRE research group for valuable comments for the article manuscript.

## Author Contributions

**Conceptualization:** Elina Late, Jaana Kekäläinen.

**Data curation:** Elina Late, Jaana Kekäläinen.

**Formal analysis:** Elina Late, Jaana Kekäläinen.

**Investigation:** Elina Late, Jaana Kekäläinen.

**Methodology:** Elina Late, Jaana Kekäläinen.

**Project administration:** Elina Late, Jaana Kekäläinen.

**Resources:** Elina Late, Jaana Kekäläinen.

**Software:** Elina Late, Jaana Kekäläinen.

**Validation:** Elina Late, Jaana Kekäläinen.

**Visualization:** Elina Late, Jaana Kekäläinen.

**Writing – original draft:** Elina Late, Jaana Kekäläinen.

**Writing – review & editing:** Elina Late, Jaana Kekäläinen.

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
