## [Decision Letter · Decision Letter 0]

30 Jan 2020

PONE-D-19-34995

Use and users of social science research data archive

PLOS ONE

Dear Dr. Late,

Thank you for submitting your manuscript to PLOS ONE. After careful consideration, we feel that it has merit but does not fully meet PLOS ONE’s publication criteria as it currently stands. Therefore, we invite you to submit a revised version of the manuscript that addresses the points raised during the review process.

We would appreciate receiving your revised manuscript by Mar 15 2020 11:59PM. To enhance the reproducibility of your results, we recommend that if applicable you deposit your laboratory protocols in protocols.io, where a protocol can be assigned its own identifier (DOI) such that it can be cited independently in the future. For instructions see: http://journals.plos.org/plosone/s/submission-guidelines#loc-laboratory-protocols

We look forward to receiving your revised manuscript.

Kind regards,

Shailesh Kumar

Academic Editor

PLOS ONE

Journal Requirements:

2. We note that you have stated that you will provide repository information for your study once permission for data sharing is recieved. Should your manuscript be accepted for publication, we will hold it until you provide the relevant accession numbers or DOIs necessary to access your data. If you wish to make changes to your Data Availability statement, please describe these changes in your cover letter and we will update your Data Availability statement to reflect the information you provide.

3. Please include a copy of Table 1 which you refer to in your text on page 10 and 11.

Reviewers' comments:

Reviewer's Responses to Questions

**Comments to the Author**

1. Is the manuscript technically sound, and do the data support the conclusions?

Reviewer #1: Yes

Reviewer #2: Yes

2. Has the statistical analysis been performed appropriately and rigorously? 

Reviewer #1: Yes

Reviewer #2: Yes

3. Have the authors made all data underlying the findings in their manuscript fully available?

Reviewer #1: Yes

Reviewer #2: No

4. Is the manuscript presented in an intelligible fashion and written in standard English?

Reviewer #1: Yes

Reviewer #2: Yes

5. Review Comments to the Author

Reviewer #1: 1. Author need to justify the statement in line number 122, “the reuse of quantitative data is far more typical than reuse of qualitative data”.

2. Author need to justify as to why certain data download involves registration while others do not.

3. Authors have used two contradictory statements. In line number 177, it mentioned as “the number of cases (N) varies in our data between 7566-10346”. However, in line number 178, 179, it is mentioned as, “in total data contains 10346 cases”. How many total data cases are there in FSD database?

4. Table 1 is missing from the manuscript. Authors are required to include the table in the manuscript

Reviewer #2: The study on the use and users of a social science research data archive (FSD) provides an interesting insight to the practices and purposes of reusing open data from social science research. By exploring the different types of data, disciplines, background of users, and their purposes, this research throws light on the significance of open data, the possibilities of sharing such data, its limitations, awareness concerning citation and important ethical issues. (On page 6, line 126, the sentence seems to be incomplete - "improper manipulation of". Kindly check it.) Overall, the findings provide comprehensive answers to the research questions posed at the beginning of the study, and the data referred to, clearly supports the conclusion, highlighting on its active usage in the field of education, not only among Finnish scholars, but users located in and outside Europe as well. Due to the lack of permission from the data donator, the complete data used for this research has not been shared publicly by the authors and will be done once the permission to do so is granted.

6. PLOS authors have the option to publish the peer review history of their article (what does this mean?). If published, this will include your full peer review and any attached files.

Reviewer #1: No

Reviewer #2: No

---

## [Author Response · Author response to Decision Letter 0]

14 Mar 2020

Dear Editor-in-Chief,

we would like to thank the two referees for their valuable comments to our article manuscript. Next, we will answer for the comments from the referees. Corrections to the text can be seen also from the file “Revised manuscript with track changes”. 

1. Author need to justify the statement in line number 122, “the reuse of quantitative data is far more typical than reuse of qualitative data”. 

It is true, that there is a lack of research of the share of reusing quantitative and qualitative datasets. However, it is known that the availability of quantitative datasets is greater. We edited the sentence:

“The reuse of quantitative data is probably more common than reuse of qualitative data because the number of opened quantitative datasets is greater [51] and metadata for quantitative data are easier to produce. Nevertheless, studies on the reuse of qualitative data in social sciences are quite numerous.”

2. Author need to justify as to why certain data download involves registration while others do not. 

In FSD the depositor of the data can decide the availability category of the archived datasets. We added a sentence in lines 167-168 for clarification: 

“The depositor of the data can decide on which terms the dataset can be downloaded.”

3. Authors have used two contradictory statements. In line number 177, it mentioned as “the number of cases (N) varies in our data between 7566-10346”. However, in line number 178, 179, it is mentioned as, “in total data contains 10346 cases”. How many total data cases are there in FSD database?

The number of cases varies between 7566-10346. We removed the sentence from the text: 

“In total data contains 10346 cases.”

4. Table 1 is missing from the manuscript. Authors are required to include the table in the manuscript.

There was a mistake in the numbering of the tables. Table numbering is now correct. 

5. In line 128 we corrected the incomplete sentence “improper manipulation of data.”

6. Data availability

We have archived our data to Finnish social science research data archive. However, the dataset have not been published yet. It will happen during the next month. Dataset will be available for research and teaching purposes at https://www.fsd.tuni.fi/

With kind regards,

Elina Late & Jaana Kekäläinen

---

## [Decision Letter · Decision Letter 1]

6 May 2020

Use and users of social science research data archive

PONE-D-19-34995R1

Dear Dr. Late,

We are pleased to inform you that your manuscript has been judged scientifically suitable for publication and will be formally accepted for publication once it complies with all outstanding technical requirements.

With kind regards,

Shailesh Kumar

Academic Editor

PLOS ONE

Additional Editor Comments (optional):

Dear Authors, Your submission is accepted for publication.

Reviewers' comments:

Reviewer's Responses to Questions

**Comments to the Author**

1. If the authors have adequately addressed your comments raised in a previous round of review and you feel that this manuscript is now acceptable for publication, you may indicate that here to bypass the “Comments to the Author” section, enter your conflict of interest statement in the “Confidential to Editor” section, and submit your "Accept" recommendation.

Reviewer #1: (No Response)

2. Is the manuscript technically sound, and do the data support the conclusions?

Reviewer #1: (No Response)

3. Has the statistical analysis been performed appropriately and rigorously? 

Reviewer #1: (No Response)

4. Have the authors made all data underlying the findings in their manuscript fully available?

Reviewer #1: (No Response)

5. Is the manuscript presented in an intelligible fashion and written in standard English?

Reviewer #1: (No Response)

6. Review Comments to the Author

Reviewer #1: (No Response)

7. PLOS authors have the option to publish the peer review history of their article (what does this mean?). If published, this will include your full peer review and any attached files.

Reviewer #1: No

---

## [Editor Report · Acceptance letter]

24 Jul 2020

PONE-D-19-34995R1 

Use and users of a social science research data archive 

Dear Dr. Late:

I'm pleased to inform you that your manuscript has been deemed suitable for publication in PLOS ONE. Congratulations! Your manuscript is now with our production department. 

Kind regards, 

on behalf of

Dr. Shailesh Kumar 

Academic Editor

PLOS ONE